# Lociq provides a loci-seeking approach for enhanced plasmid subtyping and structural characterization

Lucas Harrison [1]✉, Shaohua Zhao[1], Cong Li[1], Patrick F. McDermott[1], Gregory H. Tyson[1] & Errol Strain[1]

Antimicrobial resistance (AMR) monitoring for public health is relying more on whole genome sequencing to characterize and compare resistant strains. This requires new approaches to describe and track AMR that take full advantage of the detailed data provided by genomic technologies. The plasmid-mediated transfer of AMR genes is a primary concern for AMR monitoring because plasmid rearrangement events can integrate new AMR genes into the plasmid backbone or promote hybridization of multiple plasmids. To better monitor plasmid evolution and dissemination, we developed the Lociq subtyping method to classify plasmids by variations in the sequence and arrangement of core plasmid genetic elements. Subtyping with Lociq provides an alpha-numeric nomenclature that can be used to denominate plasmid population diversity and characterize the relevant features of individual plasmids. Here we demonstrate how Lociq generates typing schema to track and characterize the origin, evolution and epidemiology of multidrug resistant plasmids.

[1] Center for Veterinary Medicine, U.S. Food and Drug Administration, Laurel, MD, USA. ✉email: lucas.harrison@fda.hhs.gov

Plasmid-mediated antimicrobial resistance (AMR) allows bacteria to resist exposure to every major class of antibiotics. Transferrable plasmids can disseminate AMR genes between bacterial genera and facilitate the spread of antimicrobial-resistant pathogens among host species[1]. The National Antimicrobial Resistance Monitoring System (NARMS) recognizes plasmid-mediated AMR as a key threat to human, companion animal and food animal health. However, plasmids that encode for AMR genes are prone to genetic recombination events[2]. This capacity for genetic remodeling contributes to the great sequence diversity seen among plasmids and confounds current efforts to track and characterize these clinically relevant molecules[3–8].

The most common plasmid typing method categorizes plasmids by a single conserved region on the plasmid replicon[9]. This method, plasmid incompatibility group typing (Inc typing), uses a PCR based replicon typing approach and emerged from research on the effects of plasmid replicon pairs and plasmid replication efficiency[10]. Plasmid combinations that result in decreased replication efficiency when concurrently occupying the same cell are classified within the same incompatibility group. This criterion was well-suited for analysis with contemporaneous molecular or in silico methods because it only required identification of a single target[11]. However, this reliance on a single genetic target does not address the great sequence diversity present among plasmids within a single Inc group and often does not detect hybrid plasmids[12]. This shortfall of using a single target for plasmid typing is apparent when the plasmid contains multiple replicon sequences[13,14].

A complementary method to plasmid Inc typing known as MOB typing categorizes plasmids by the sequence of their relaxase protein[15,16]. The relaxase protein is an essential component in mobilizable plasmids that binds to the plasmid origin of transfer, introduces a single stranded nick and facilitates the transfer of the single plasmid strand to the bacterial plasmid secretion system[17]. Relaxase proteins have been phylogenetically grouped into six MOB families and plasmids are assigned to a MOB group based on the relaxase protein sequence[16]. Unfortunately, MOB typing methods are limited in their ability to categorize non-mobilizable plasmids and like Inc typing are based on only a single target.

One promising typing approach classifies plasmids by their average nucleotide identity[18]. This approach has a notable advantage over other typing schema because it uses the entire plasmid sequence to identify plasmid taxonomic units (PTUs) instead of using a single target. The PTU method identifies conserved taxonomic units using a sequence-length dependent comparison between plasmids. One of the main advantages of this method is that it classifies plasmids independent from any predicted phenotypic trait or function. This sequence-based approach has shown strong associations between PTU group and bacterial host specificity[19]. However, this approach does have limitations. First, because this method makes length-based comparisons between plasmids it is possible to miss regions of sequence similarity in smaller plasmids when they fall below the method's cutoff threshold. Second, the naming schema of the PTU system is independent of other typing systems, complicating comparisons to historical plasmid data. Finally, similar to other average nucleotide identity clustering methods, this method does not take into account variations in the plasmid structure resulting from recombination events. These limitations hinder the ability to make detailed comparisons between plasmids using the PTU designation alone.

Plasmid multilocus sequence typing (PMLST) addresses some of the challenges of Inc, MOB and average nucleotide identity typing methods. Schema that contain more than one target for plasmid typing are able to account for a greater degree of sequence diversity within a plasmid type[3]. Unlike the MOB methods, PMLST is able to categorize non-mobilizable plasmids as well. PMLST methods are compatible with existing plasmid typing nomenclature and the typing loci are defined sequences that can be used in downstream analysis. The IncA/C[3], IncF[4], IncHI[6], IncH2[5], IncI1[7] and IncN[8] PMLST schema contain 2-6 typing loci each and have contributed greatly to the understanding of plasmid sequence diversity. The IncA/C PMLST schema is used to differentiate the plasmids of the IncC plasmid type. The IncC plasmids are commonly associated with the carriage of clinically-relevant antimicrobial resistance genes and contribute to the spread of the multi-drug resistant phenotype[20]. Core genome plasmid multilocus sequence typing (cgPMLST) expands on PMLST methods further by identifying the genes essential for plasmid maintenance and using them as sequence typing targets[3]. This method has been applied to IncA/C plasmids to increase the number of typing loci to 28. However, while more targets are used for PMLST- based plasmid classification, they only represent a small percentage of the entire plasmid sequence and provide little information on structural differences between plasmids.

One factor that has hindered the progress of sequence-based plasmid typing systems is the difficulty of assembling plasmids from short read sequencing data[21]. However, as long-read sequencing technologies become more accessible, more closed plasmid assemblies are available to researchers. Closed plasmid assemblies offer two main advantages over gapped, or draft, plasmid assemblies. First, closed plasmid assemblies account for every nucleotide on the plasmid molecule. This provides a full accounting of all the coding and intergenic regions on the plasmid. The second advantage closed assemblies provide is the ability to determine which sequences are missing from a plasmid. For comparison, draft assemblies do not contain the entire plasmid sequence and cannot be used to determine if a given sequence is missing. Finally, closed assemblies can be used to identify the relative position of any genetic element on the plasmid. This attribute is useful in epidemiological operations such as antimicrobial resistance monitoring where the proximity of an AMR gene to an insertion sequence or transposon can help assess the risk of gene transfer.

These three attributes of closed plasmid assemblies are ideal factors for plasmid typing. First, the ability to account for every nucleotide on the plasmid increases the likelihood of identifying common sequences shared among different plasmids. Second, the ability to equate absence of sequence in an assembly to absence of sequence in the cognate plasmid allows for plasmid classification methods based on the presence or absence of genetic elements. Finally, analyzing the relative position of each genetic element on the plasmid can account for differences in the plasmid structure resulting from plasmid recombination and insertion events.

Here we present a plasmid subtyping method that uses closed plasmid assemblies to identify the conserved sequences and patterns of loci found among plasmids of a given plasmid type. In this paper, we propose to subtype plasmids of the IncC plasmid type as a demonstration of the Lociq method. We chose the IncC plasmids, not only because of their role in the transmission of AMR genes, but also because we can compare results of the Lociq method to the PMLST and cgPMLST profiles of this well-characterized plasmid type. By identifying these conserved genetic elements and patterns, we aim to develop a scalable approach to plasmid classification that allows the user to first identify large families of plasmids and then apply additional typing criteria to differentiate between individual plasmids. The purpose of this paper is to introduce the plasmid subtyping method, demonstrate its ability to subtype IncC plasmids, compare it to existing plasmid typing methods, and show how the results of the subtyping method can be used to facilitate

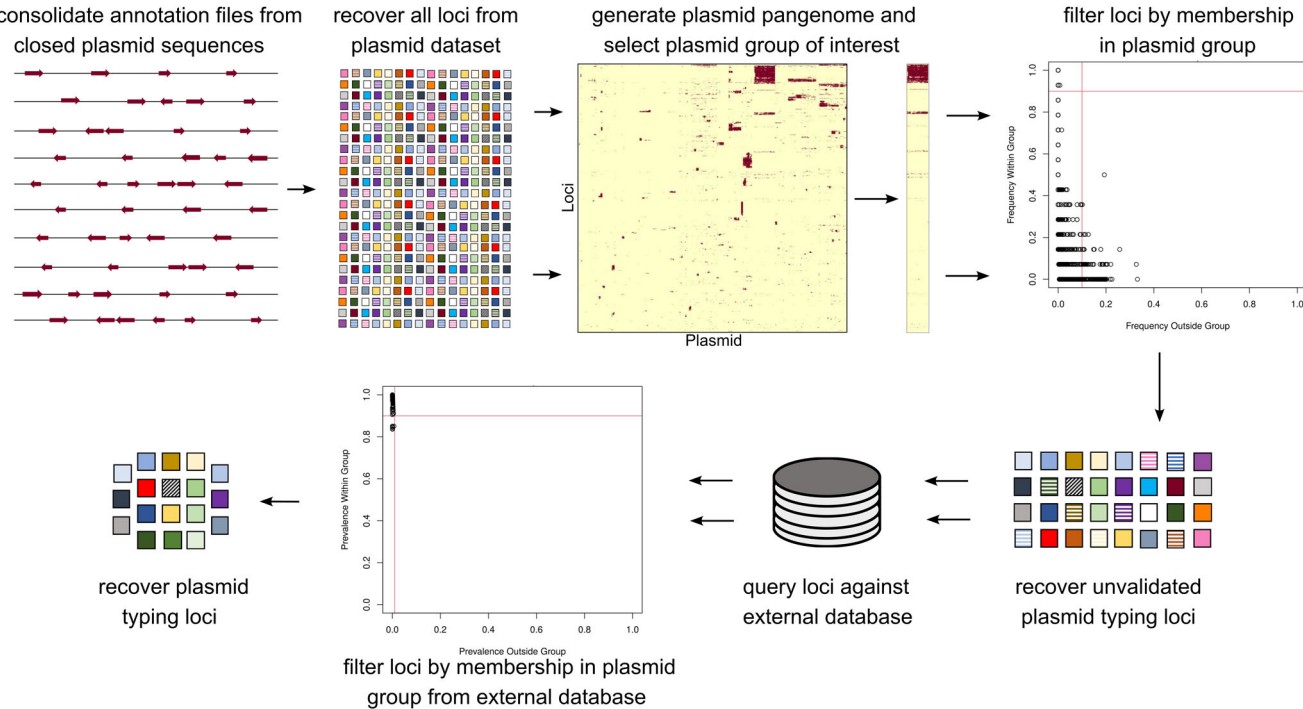

**Fig. 1 Lociq workflow for the identification of plasmid typing Loci.** Overview of the typing loci identification process of the Lociq method.

research in plasmid biology, which has the potential to enhance pathogen surveillance for public health.

## Results

We demonstrated the utility of the Lociq plasmid typing method by performing an analysis of closed plasmid assemblies and generating subtyping definitions for the IncC plasmids. Identification of the typing loci was performed by using the Roary and piggy programs to define the pangenome of 459 closed plasmid sequences[22,23]. Prevalence thresholds were used to determine which pangenomic loci were indicative of and exclusive to a given plasmid type. Finally, the candidate typing loci were validated against an external database (Fig. 1). We then compared the Lociq typing method results to Inc, MOB and PTU typing methods, as well as PMLST and cgMLST subtyping methods. Finally, we demonstrated how the Lociq method organizes the results to facilitate downstream analyses.

**Plasmid subtyping method.** The full dataset of *Salmonella* and *E. coli* isolates contained 459 closed plasmid assemblies and 46 plasmid Inc types. These 46 plasmid types were represented by 398 plasmids and the remaining 61 plasmids did not belong to any plasmid Inc group. The combined pangenome for all 459 plasmids contained 6726 unique coding and intergenic regions, as generated by the Roary & piggy programs. These 6726 genetic elements were the library of plasmid loci found among our plasmids. The pangenome was analyzed as a binary presence/absence matrix in R where plasmids were grouped by the similarity of their loci profiles accounting for both the coding and intergenic regions. This grouping was performed first by computing a distance matrix of the binary matrix data, then clustering with the hclust function using complete linkage. The resulting presence absence matrix was used for downstream subtyping of the Inc group plasmid typing schema (Fig. 2).

Next, we identified the IncC cluster on the presence-absence matrix and selected the loci indicative of and exclusive to IncC

plasmids. Identification of the IncC plasmids revealed that the loci composition of IncC plasmids is not uniform and only a subset of loci is shared among the IncC plasmids (Fig. 2). Next, we identified the loci indicative of and selective for IncC plasmids by comparing the prevalence of each 6726 loci among IncC plasmids to their prevalence in non-IncC plasmids. Seventy-five loci were present in >90% of the IncC plasmids and fewer than 10% of the non-IncC plasmids (Supplementary Fig. 1). This initial set of IncC typing loci contained 59 coding and 16 intergenic regions.

Following the initial identification of typing loci, we evaluated the prevalence of the loci against plasmids in an external database. The purpose of this analysis was to reduce the bias in loci selection that may be introduced if the initial dataset was not representative of the broader plasmid population. For example, in this demonstration, all plasmids were harvested from *Salmonella* and *E. coli* strains that were isolated from retail meats or food animal cecal samples. The plasmid data set did not contain plasmids harvested from other genera of bacteria and none of the bacteria were isolated from clinical or environmental sources. To address this, we evaluated the prevalence of the 75 loci among the 34,513 plasmids of the PLSDB v.2021_06_23_v2 database[24]. We compared the prevalence of typing loci between IncC and non-IncC plasmids in the database. Seventy-two of the seventy-five IncC typing loci met the two criteria of being present in > 90% of the plasmids that contained at least one typing locus and being present in < 1% of plasmids without a typing locus in the PLSDB database. The resulting complement of 72 IncC typing loci accounted for 40,091 bp and contained 58 coding regions and 14 intergenic regions. Further, a 90% prevalence of loci threshold was sufficient to identify all 534 IncC plasmids in the PLSDB database.

In the next stage of this plasmid subtyping demonstration, we identified the patterns of contiguous plasmid loci that were conserved among the IncC plasmids in the PLSDB database. These conserved contiguous regions were identified as fragments of the plasmid backbone. The fragment analysis that allowed for

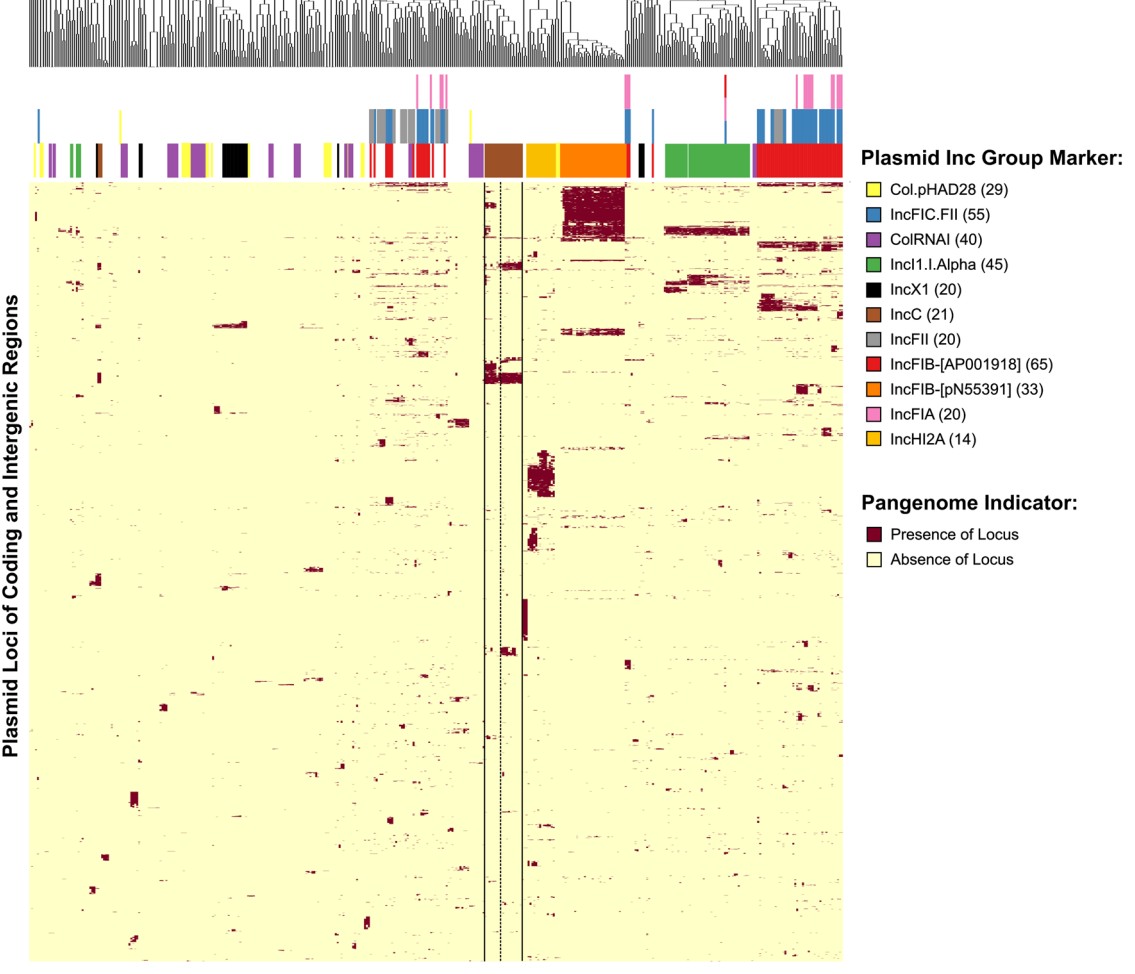

**Plasmid Inc Group Marker:**

- Col.pHAD28 (29)
- IncFIC.FII (55)
- ColRNAI (40)
- IncI1.I.Alpha (45)
- IncX1 (20)
- IncC (21)
- IncFII (20)
- IncFIB-[AP001918] (65)
- IncFIB-[pN55391] (33)
- IncFIA (20)
- IncHI2A (14)

**Pangenome Indicator:**

- Presence of Locus
- Absence of Locus

**459 Closed Plasmid Sequences from NARMS Sampling**

**Fig. 2 Pangenome of 459 closed plasmid sequences.** Presence Absence matrix of loci in the plasmid dataset. Plasmids are arranged along the x-axis while the loci of coding and intergenic regions are organized along the y-axis. Dark red color indicates the presence of a locus in a given plasmid. Plasmid Inc group assignments of the most common Inc types are visually located above each plasmid as well as in tabular format in Supplementary Data 1. Multiple Inc group markers in a single plasmid are stacked vertically. The larger cluster of IncC plasmids is delineated by solid lines while the two IncC subclusters are subdivided by a dotted line.

no greater than 500 bp between neighboring loci revealed that the IncC plasmids contained 8 conserved plasmid fragments. These fragments contained 2–31 typing loci (Fig. 3) and the loci sequences on the fragments ranged in size from 236 bp to 13,836 bp (Supplementary Data 2). The mean correlation coefficient for the 31 loci on the largest plasmid fragment was 0.989 (Supplementary Data 3). Fragment 6 had the lowest mean correlation coefficient among its 4 loci with an R-value of 0.916. The 417 bp PlasmidFinder IncC marker was contained within a 1066 bp locus found on plasmid fragment 1. This fragment contained 8 loci with a mean R-value among its loci of 0.942.

In the final stage of our demonstration of IncC plasmid analysis with the Lociq method, we used the sequence and position of the typing loci to characterize all plasmids from the external database that contained at least 1 IncC typing locus (Supplementary Fig. 2). Plasmid characterization was performed by assigning a numeric identifier to each unique pattern of sequence type, fragment type and loci type (Fig. 4). The plasmid sequence type was defined by the complement of plasmid alleles in the plasmid, regardless of their position. The plasmid fragment type was determined by how the plasmid fragments were ordered along the plasmid, relative to a semi-conservative starting locus. The plasmid loci type was determined by rearranging the plasmid fragments in ascending

order of their numeric identifier and recovering the arrangement of loci from the re-ordered plasmid fragments. This temporary rearrangement of plasmid fragments for loci typing allows the loci type to be independent of the fragment type.

In addition to the 534 IncC plasmids in the database of 34,513 plasmids, the analysis identified 31 IncC hybrid plasmids that contained at least 1 of the IncC typing loci. The 534 IncC plasmids were then subdivided into unique patterns of 52 fragment types, 260 loci types and 388 sequence types. There were 397 unique combinations of fragment type, loci type and sequence type represented among the 534 IncC plasmids (Supplementary Data 4). Further, the addition of the interfragment distance values to the subtyping criteria increased the number of unique combinations to 515. As a result, the 534 IncC plasmids could be divided into 515 unique combinations of fragment type, loci type, sequence type and interfragment distances.

The Lociq plasmid subtyping method includes features for analysis of the results. First, the results can be evaluated in a web-browser using an R-shiny application[25]. This application allows the user to compare plasmids by generating a graphical map (Fig. 5) of each plasmid in the database (Supplementary Fig. 3), a report of plasmid features (Supplementary Fig. 4) and a searchable table of AMR genes that are present in the plasmid (Supplementary Fig. 5)

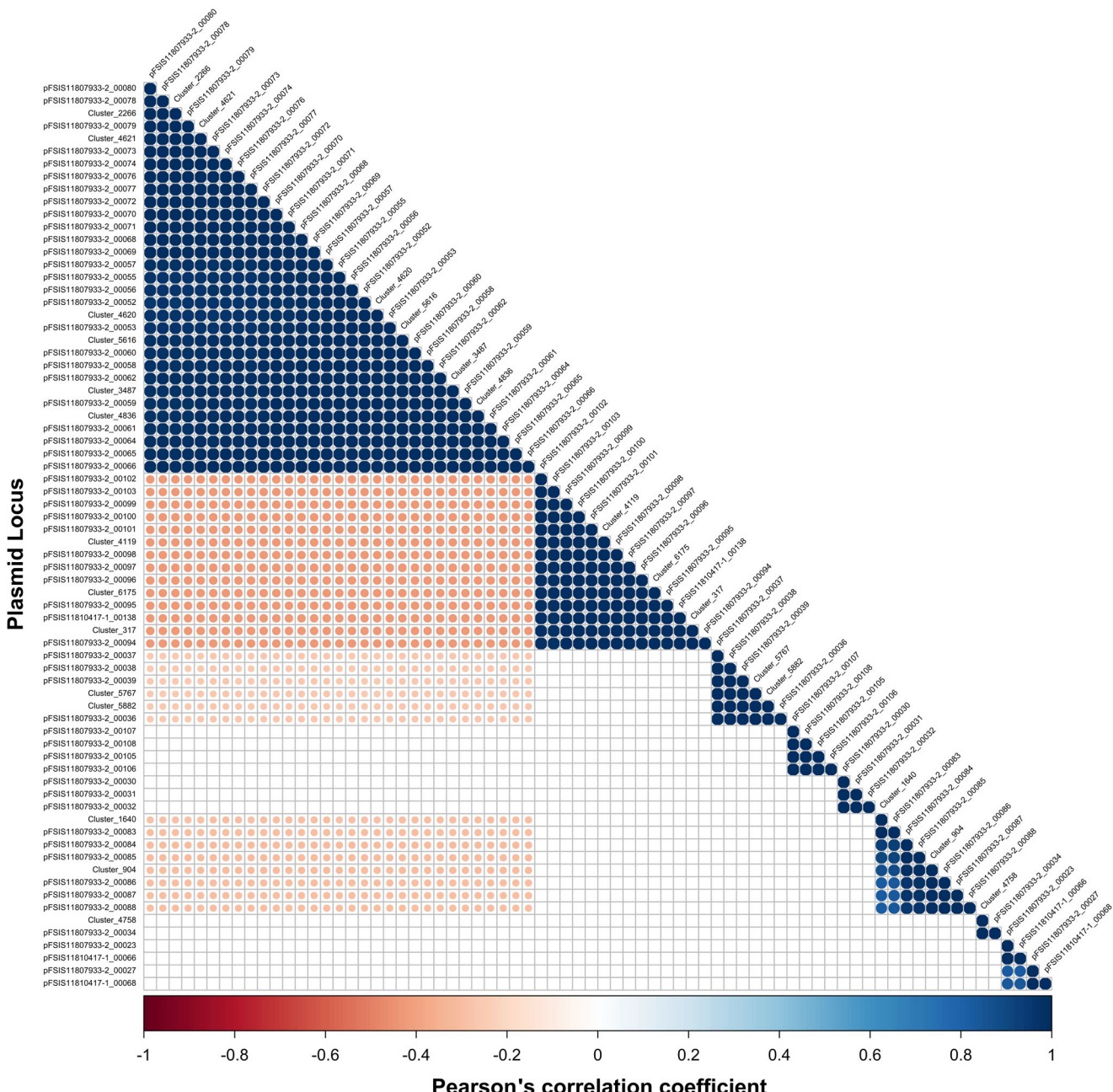

**Fig. 3 Correlation matrix and fragment assignments of IncC Plasmid Loci.** A correlogram illustrating the strength of correlation between any two loci occurring on the same contiguous region of an IncC plasmid ($n = 565$ plasmid sequences that contain IncC typing loci). The conserved fragments can be seen as dark blue circles in eight distinct triangular shapes along the diagonal. Correlation coefficients corresponding to insignificant loci interactions ($p \geq 0.05$) are represented as blank cells. Source data for the correlogram may be found in Supplementary Data 3.

and the full plasmid database (Supplementary Fig. 6). Second, this subtyping method generates a tabular typing summary of all the plasmids that were evaluated (Table 1). This summary includes the plasmid ID, plasmid typing category, fragment type, loci type, plasmid sequence type, fragment sequence types and the interfragment distances (Supplementary Data 4). Third, the method produces sequence (Supplementary Data 5) and pattern (Supplementary Data 6) definitions for downstream analysis. Finally, this subtyping method includes a script that allows the user to characterize their own plasmid sequences using the database of results generated by the Lociq method. The database will also update the plasmid typing reference database if the user's plasmid sequences contain variants in sequence or structure that were not previously accounted for.

**Comparison to existing methods**. Our subtyping method classifies plasmids by variations in loci sequence and relative position on the plasmid. We compared the total number of subtyping groups, the size of each group and the Simpson diversity index across four plasmid typing methods and the Lociq typing method to evaluate their discriminatory power (Fig. 6). The first two methods we evaluated were the MOB type and the PTU typing methods. While neither of these classification methods were designed for IncC plasmid subtyping, both are valuable alternatives to the Inc typing system. PTU classification of the plasmids was able to assign 479 of the 534 IncC plasmids to a PTU group. The largest group contained 475 plasmids and the results that were generated had a Simpson's index of diversity of 0.199. MOB typing of the 534 IncC plasmids revealed 15 MOB types,

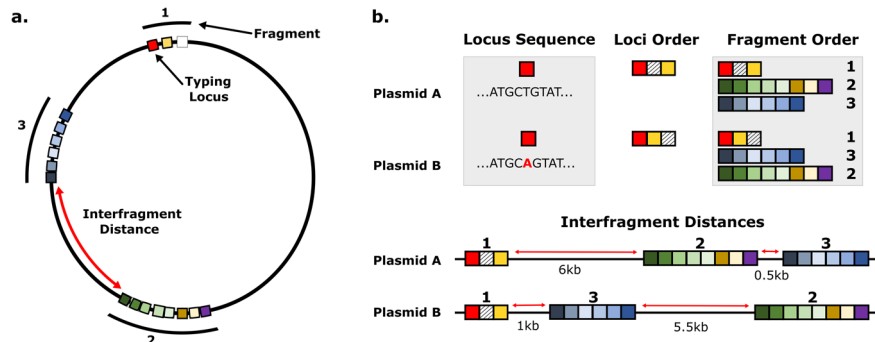

**Fig. 4 Metrics for plasmid typing.** Endpoints evaluated in the Lociq plasmid typing method (**a**). Boxes represent plasmid loci while the numbered clusters of loci correspond to plasmid fragments. Examples of how the endpoints can be used to differentiate between two example plasmids A and B (**b**) using the sequence of plasmid loci to determine plasmid sequence type, order of the plasmid loci to determine loci type, order of the plasmid fragments to determine fragment type or the distances between the plasmid fragments as a metric for interfragment distances.

**Fig. 5 Lociq characterization of IncC plasmids.** A graphical representation of nine IncC plasmids generated by the Lociq companion application. The numbered black bars represent plasmid fragments, red bars represent AMR genes and yellow bars represent stress-tolerance genes. Strand orientation is in relation to the plasmid indexing locus and forward orientation is represented by gene presence above the sequence line.

**Table 1 Sample Results from IncC Plasmid Subtyping.**

| Plasmid | Fragment Pattern | Loci Pattern | Sequence Type | Interfragment Distance (bp) |
|---|---|---|---|---|
| AP022381.1 | 25 | 3 | 376 | 2942, 14165, 870, 115695, 17808, 7594 |
| AP022385.1 | 25 | 3 | 376 | 2942, 14165, 870, 84976, 17808, 7594 |
| AP024844.1 | 25 | 2 | 381 | 2942, 1030, 870, 103967, 17766, 3933, 3133 |
| CP063757.1 | 27 | 134 | 147 | 1226, 1030, 870, 40258, 2955, 1100, 13770 |
| NZ_CP045517.1 | 24 | 205 | 126 | 1226, 1030, 870, 15107, 637, 688, 829, 25736 |
| NZ_CP048384.1 | 30 | 77 | 226 | 1226, 1030, 870, 130822, 829, 688, 637, 11581 |
| NZ_MH995506.1 | 45 | 255 | 319 | 637, 11153, 1142, 1226, 1030, 870, 80076, 829 |
| NZ_LT985224.1 | 30 | 151 | 75 | 1226, 1030, 11687, 870, 70622, 829, 688, 637, 27441 |
| NZ_CP065463.1 | 30 | 79 | 239 | 6378, 870, 71929, 829, 688, 637, 25439 |

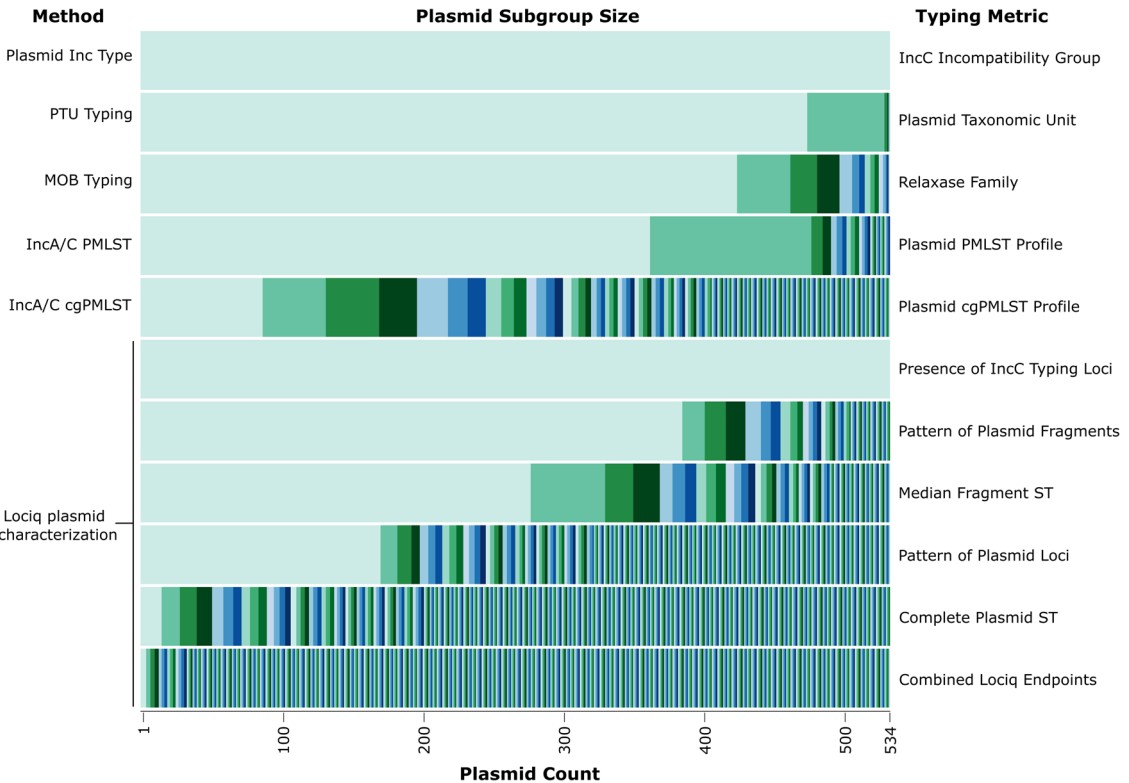

**Fig. 6 Comparison of plasmid typing methods.** Classification comparison of 6 plasmid typing methods. A stacked barplot comparing the size of each subgroup that was identified from the initial dataset of 534 IncC plasmids across 6 typing methods. Subgroups for each method are arranged in decreasing size from left to right and source data are available in Supplementary Data 7.

the largest of which contained 425 plasmids. MOB typing of this dataset generated a Simpson's index of diversity of 0.359. The next two methods we evaluated were specifically designed to subtype IncC plasmids and showed greater ability to differentiate between plasmids. The first of these methods was the 5 loci IncA/C PMLST schema which produced 28 groups. The largest group classified by this method contained 363 plasmids and the diversity index for this method was 0.492. The final comparator typing method was the 28 loci IncA/C cgPMLST schema. There were 180 unique combinations of IncA/C cgPMLST alleles represented in the dataset and the most common combination was identified in 87 plasmids. Typing with the IncA/C cgPMLST loci showed the greatest discriminatory power of all the comparator methods with a Simpson's diversity index of 0.954.

Next, we evaluated the typing schema generated in our plasmid subtyping method (Fig. 6). Structural characterization of the plasmids by the order of their fragments grouped the 534 IncC plasmids into 53 groups, the largest of which contained 386 plasmids. Fragment typing had slightly greater discriminatory

power than MOB typing, as indicated by a Simpson's diversity index of 0.475. Structural characterization of plasmids by the order of their loci classified the plasmids into 260 groups. The largest group contained 171 plasmids and the Simpson's diversity index for this schema was 0.896. This value was slightly less than the diversity index of the IncA/C cgMLST method. The plasmid classification schema that grouped plasmids by the plasmid loci sequence type that were generated in our method grouped the plasmids into 388 groups. The largest group that was produced with this schema contained 15 plasmids. This schema had the second highest Simpson's diversity index of 0.996. The final schema that we evaluated combined all the structural and sequence features that were generated in the analysis. For this aggregate schema, plasmids were evaluated by their fragment type, loci type, sequence type and the distances between their fragments. This separated the plasmids into 515 groups, and the largest group contained 4 plasmids. This schema had the greatest discriminatory power with a Simpson's diversity index >0.999.

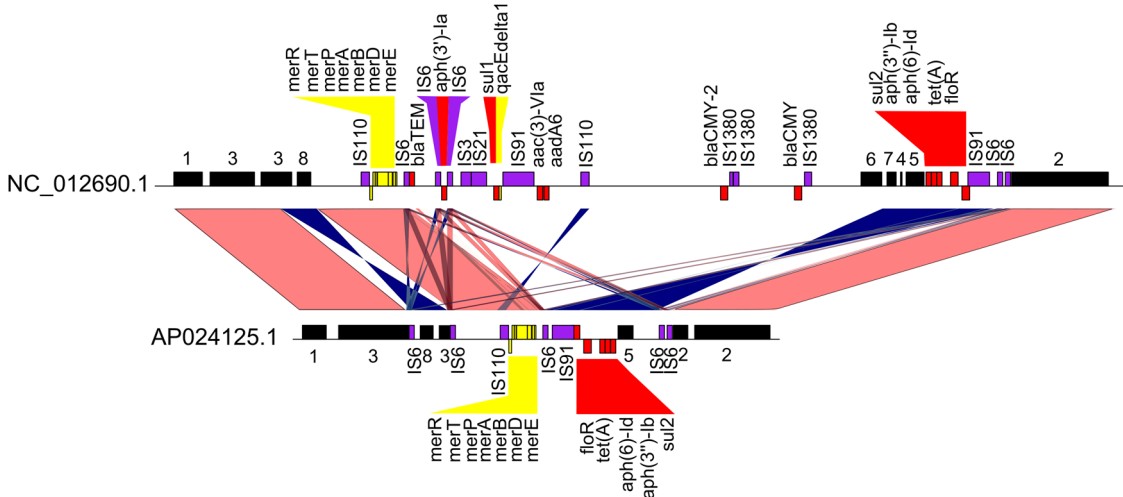

**Fig. 7 Alignment of IncC plasmids with insertion sequence custom annotations.** Alignment of two IncC plasmids with Insertion Sequence (IS) element annotations and sequence alignments added. Black bars represent IncC plasmid fragments, red bars represent AMR genes, yellow bars represent stress tolerance genes and purple bars represent IS elements. Light red alignments depict conserved regions in the same orientation in both plasmids while dark blue alignments show sequence inversions. Notice the two prominent sequence inversions in AP024125.1 are immediately flanked by IS6 elements.

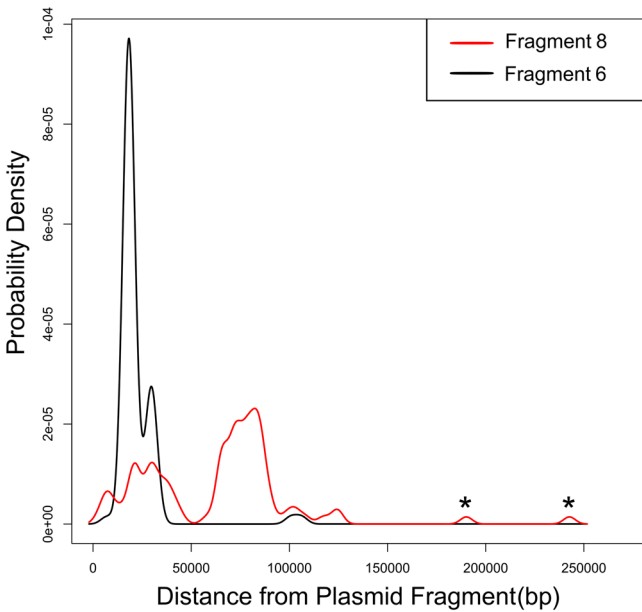

**Fig. 8 Location of *bla*CMY-2 relative to IncC plasmid fragments.** A density plot of the position of *bla*CMY-2 upstream of IncC plasmid fragment 6 and downstream of IncC plasmid fragment 8. The two small peaks near 190,000 bp and 240,000 bp correspond to plasmids NZ_CP019001.1 and NZ_CP028804.1, respectively. Source data are available in Supplementary Data 8.

**Downstream analyses of hybrid plasmids and custom annotations.** Typing plasmids using the Lociq method allows us to standardize the language surrounding plasmid feature diversity. Here, we present four demonstrations of how the Lociq results can be applied to downstream analyses. First, because the plasmid typing process indexes the plasmids to a common starting point, the data are organized to facilitate the incorporation of custom feature annotations. This can aid the identification of hybrid plasmids containing elements from multiple plasmid types, such as the hybrid plasmid NZ_CP028197.1 that contains both IncC and IncHI2A elements (Supplementary Fig. 7). The Lociq method can also be used to identify custom features such as IS elements to indicate potential sites of plasmid recombination. A comparison

of plasmids NC_012690.1 and AP024125.1 illustrates the proximity of IS elements to AMR and heavy metal resistance genes (Fig. 7). In addition, two inverted sequence regions are flanked by IS6 elements in AP024125.1 relative to NC_012690.1.

**Downstream analysis of AMR positions in a dataset**. As a second downstream analysis, we can leverage the AMR gene location data to identify trends in gene position among the plasmid dataset. We analyzed the location of *bla*CMY-2 among our IncC plasmids. Of the 117 plasmids that encoded for *bla*CMY-2, 93 plasmids bore the gene downstream of IncC fragment 8 and upstream of IncC fragment 6. All but 2 of the *bla*CMY-2 genes in this subset were located in a range that peaked at 28 kb upstream of IncC fragment 6 (Fig. 8). The *bla*CMY-2 genes were located in two ranges downstream of IncC fragment 8. One range peaked at 30 kb downstream of fragment 8 and the other at 80 kb. Two *bla*CMY-2 genes were found outside of these ranges: 1 was identified 242,666 bp downstream of fragment 8 in plasmid NZ_CP028804.1 and the other 190,156 bp downstream of fragment 8 in plasmid NZ_CP019001.1. The shift in the location of *bla*CMY-2 in both cases was associated with a potential insertion event upstream of the gene. Upstream of *bla*CMY-2 in NZ_CP028804.1 is a region that contains genes associated with resistance to silver, copper and arsenic as well as heat shock tolerance as well as the genetic markers for the plasmid replicons IncFIA_1(AP001918), IncFIC(FII)_1(AP001918) and IncFII_1(AY458016). Similarly, upstream of *bla*CMY-2 in NZ_CP028804.1 is a region encoding for the *iucA, iucB, iucC, iucD, iutA* virulence genes and the genetic markers for plasmid replicons IncFIB(K)_1(JN233704) and IncFII(K)_1(CP000648). The presence of multiple plasmid replicons combined with the relative position of *bla*CMY-2 from the IncC plasmid fragments indicates these two plasmids are the result of a recombination event between an IncC plasmid and a plasmid of the IncF family of plasmid groups. The Lociq typing method records the gene position data for all AMR and user-defined accessory genes and as a result, this gene location analysis can be performed for any gene represented in the plasmid dataset.

**Lociq typing of draft assemblies**. Draft plasmid assemblies can be analyzed by using the allele definitions of the Lociq results. The Lociq program cannot analyze draft assemblies for structural variations of loci or fragment order, but it can perform plasmid

MLST to identify which plasmids in the Lociq results most closely match the draft assembly. To demonstrate this plasmid typing function, we queried the NCBI database for draft assemblies containing IncC plasmid sequence and filtered the results for assemblies generated from short reads. From this, we selected the whole genome shotgun sequencing record for *Klebsiella pneumoniae* K184 (JAANYS000000000) that contained 1,477 contigs. A BLAST query of the IncC typing loci against the 6.7 Mb draft assembly identified 62 IncC typing loci in the sequence. Thirty five of the 62 loci were partial matches that either occurred at the end of a contig (Supplementary Data 9) or aligned to complementary ends of two separate contigs (Supplementary Data 10). Of the remaining 27 loci, 22 matched known alleles in the Lociq results (Supplementary Data 11). Our analysis revealed that this grouping of 22 alleles was conserved among 86 plasmids in our dataset of 534 IncC plasmids. This subset of 86 IncC plasmids represented the closest matches to the plasmid identified in the whole genome shotgun sequencing assembly based on our typing method.

**Analysis of plasmids in a clinical setting**. The final demonstration shows how subtyping with the Lociq method can aid in tracking the evolution of a plasmid in a clinical setting. To do this, we used the Lociq method to visualize the results of a study in a major hospital in Taiwan that tracked the transmission of $bla_{OXA-48}$ from a plasmid to a *K. pneumoniae* chromosome over a three-year period[26]. During this time, an accessory IncC plasmid that was retained among the *K. pneumoniae* strains had lost ~20 kb of sequence containing 9 AMR genes. The study closed the sequences of 4 IncC plasmids that were recovered from isolates in the blood of a patient suffering from bacteremia, urine of two patients suffering from urinary tract infections and pus from a patient suffering from pneumonia. Analysis of the 4 IncC plasmids revealed that all belonged to the IncC Lociq sequence type 74 (IncC Lociq ST74) and the loci and fragment patterns were identical among all four plasmids (Fig. 9). However, the interfragment distances and arrangement of AMR genes among the plasmids differed, indicating that each of the plasmids that was recovered represented a different stage in the evolution of the plasmid at the hospital. The primary study indicated that the first

stage of plasmid evolution was observed between the plasmids NZ_CP040034.1 and NZ_CP040029.1 that were isolated in the first year of the sample period. These plasmids showed an inversion of a ~ 20 kb resistance cassette containing *erm(42)-* $bla_{TEM-31}$*-rmtb1-tet(G)-floR2-sul1-qacEdelta1-aadA2-dfrA12* that was located between IncC fragments 4 & 2. The next step in plasmid evolution indicated in the primary study was observed in plasmids recovered later in the sampling period. These plasmids showed a reduction in size due to the loss of an overlapping resistance cassette containing *aac(3)-IId-dfrA12-aadA2-qacE-delta1-sul1-floR2-tet(G)-rmtb1-blaTEM-31* but leaving *erm(42)* in the plasmid. The proposed final step was the loss of $bla_{CTX-M-14}$ that was embedded between two sections of IncC fragment 3. This quick analysis revealed that even though the plasmids were identical in sequence type, loci pattern and fragment pattern, the difference in interfragment distance showed that the plasmids were not identical. Further, the fragments of the Lociq typing method provided common reference point among the plasmids to identify where each plasmid restructuring event had taken place.

Next, we compared the four IncC Lociq ST74 plasmids recovered from *K. pneumoniae* isolates in a Taiwanese hospital to the only five IncC Lociq ST75 plasmids in our results. These two plasmid sequence types differ by a single allele that encodes for an uncharacterized protein. Even though the four ST74 plasmids were all recovered from a single location and single species, the five ST75 plasmids were recovered from multiple species and multiple sites. The smaller two IncC Lociq ST74 plasmids shared the same loci and fragment pattern with the IncC Lociq ST75 plasmids NZ_LT985224.1 and NZ_MF150121.1, however the ST75 plasmids were recovered from *E. coli* in France and *K. pneumoniae* in Brazil, respectively (Supplementary Fig. 8). Alignment of the plasmids revealed 98% coverage and > 99% identity between NZ_CP040024.1 and NZ_LT985224.1 and 97% coverage and > 99% identity between NZ_MF150121.1 and NZ_CP040039.1. The third IncC Lociq ST74 plasmid NZ_CP040029.1 shared the same plasmid structure and AMR composition of the IncC Lociq ST75 plasmids NZ_MF150118.1 and NZ_CP028996.1, but the IncC Lociq ST75 plasmids were recovered from *P. mirabilis* in Brazil and *K. pneumoniae* in USA (Supplementary Fig. 9). Both NZ_MF150118.1 and NZ_CP028996.1 aligned to the ST74 NZ_CP040029.1 with 100% coverage and >99% identity. Finally, the fourth IncC Lociq

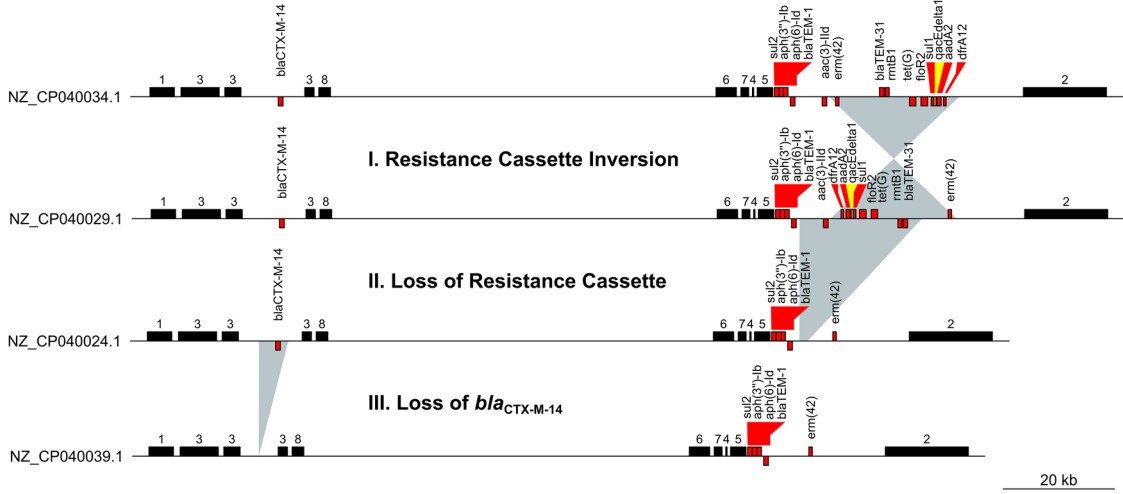

**Fig. 9 Alignment of IncC sequence type 74 plasmids.** Visual comparison of all IncC Lociq ST74 plasmids in our results. All plasmids were recovered from a single hospital in Taiwan between 2013 and 2015 and illustrate how a plasmid that is established in a single location can change over time. Shaded regions indicate differences between plasmids. The numbered black bars represent plasmid fragments, red bars represent AMR genes and yellow bars represent stress-tolerance genes. Strand orientation is in relation to the plasmid indexing locus and forward orientation is represented by gene presence above the sequence line.

ST74 plasmid NZ_CP040034.1 shared the same inverted sequence upstream of IncC plasmid fragment 2 that was observed in the IncC Lociq ST75 plasmids NZ_CP023724.1 and NZ_AP018672.1. These last two ST75 plasmids were recovered from a hypervirulent *K. pneumoniae* clinical isolate in Taiwan and a *K. pneumoniae* environmental isolate in Japan (Supplementary Fig. 10)[27]. Both ST75 plasmids shared 96% coverage and 99% identity with the ST74 plasmid, and the decreased coverage was affected by the partial loss of a resistance cassette between IncC plasmid fragment 4 and 2.

This final application of the Lociq method demonstrates how one plasmid type that was recovered solely from *K. pneumoniae* isolates from a major hospital in Taiwan was genetically similar to plasmids isolated from *K. pneumoniae*, *E. coli* and *P. mirabilis* in 4 different continents. This demonstration indicates that with the appropriate supporting epidemiological data, the results of the Lociq method can be used to support efforts to track the spread of clinically relevant plasmids.

## Discussion

We have demonstrated how the Lociq method uses closed plasmid assemblies to identify core genetic elements and structural patterns conserved among IncC plasmids. This method can be applied to a single dataset to identify typing metrics for any other plasmid group that share a core set of loci. Further, because this method can characterize plasmids that do not contain the full complement of typing loci, it is ideal for characterizing plasmids that contain elements from multiple plasmid types. This feature also allows for increased characterization of plasmids from draft assemblies where not all of the plasmid typing loci are represented in the assembled sequence. The Lociq method provides a common language to describe plasmid diversity using the endpoints of fragment pattern, loci pattern, plasmid sequence type and interfragment distances. These properties make the Lociq method a powerful tool to track and study the evolution and routes of transmission of any plasmid of interest.

The Lociq method generates multiple typing schema, each with a different discriminatory power. Typing schema with a low discriminatory power, such as the Lociq fragment type, are suited to identify larger groups of similar plasmids. The schema that accounts for all metrics of the Lociq method had the greatest discriminatory power of all the evaluated methods and is best suited to differentiate between similar plasmids. The comparator method whose metrics generated greatest discriminatory power was the IncA/C cgPMLST method. However, the IncA/C cgPMLST schema was only designed to differentiate between IncA/C plasmids while the Lociq method can theoretically be applied to characterize any plasmid type that shares a common set of core loci. Further, the cgPMLST was developed through resource intensive transposon disruption assays while the bioinformatic Lociq subtyping method can be run on a desktop computer[3].

The Lociq method adds two features that are not common in other typing methods. First, this method identifies conserved intergenic regions and codifies them as typing loci. This has the dual benefit of not only increasing the pool of plasmid loci, but also facilitating the construction of larger contiguous regions of neighboring typing loci. The second feature the Lociq method adds is an analysis of variations in the plasmid structure. Structural analysis of the arrangement of elements is relevant to plasmid typing because it can identify common recombination events in plasmids such as deletions, insertions, duplications or rearrangement events. The structural analysis also accounts for differences in the length of sequence between the plasmid fragments. Variations in interfragment distances can notify researchers not only that a recombination event occurred, but

also the region of the plasmid where the recombination event took place.

While the Lociq method increases the discriminatory power of plasmid subtyping through the addition of structural comparisons, the method does have limitations. First, the user needs to have access to a library of high-quality closed plasmid assemblies to construct their initial dataset. Second, the plasmid dataset should contain sufficient genetic diversity to represent the plasmid type of interest. The Lociq method also requires the user to input threshold values for loci selection and interfragment distance limits. The method provides graphics to help inform the user of plasmid loci distribution, but no equation to determine the optimum cutoff value for prevalence within a plasmid type is supplied with the method. Finally, even though the Lociq results demonstrated greater discriminatory power than other typing schema, increased discriminatory power is not always ideal when the objective is to identify similar members of a group. Fortunately, the Lociq method generates multiple outputs that allow the user to select the testing metric that is appropriate for their purposes. Due to these limitations, the IncC-specific typing definitions that we obtained from our sample set of foodborne pathogens are not intended to classify the full diversity of the extant IncC population. Rather, developing the plasmid typing definitions that accurately reflect the diversity of plasmid sequence and structure will require collaboration with a number of partners that represent a diverse set of isolation locations, biological compartments, host organisms and isolation dates.

In addition to the applications demonstrated earlier, this typing method has promising implications for plasmid research. First, the Lociq method can be used to characterize plasmids that are currently untyped. The initial stage of the Lociq program organizes plasmids independent of plasmid type through hierarchical clustering of loci presence/absence data. Plasmids belonging to clusters without a known plasmid type can be characterized by the Lociq method using the typing loci unique to that cluster. Second, the Lociq method can facilitate analyses between plasmid sequence and plasmid metadata. These comparisons may be made either by evaluating the sequence composition of the plasmid typing alleles, or by evaluating alleles present in subclusters of a plasmid type as was seen in the clustering of the IncC plasmids (Fig. 1). Finally, the library of typing loci may help to reconcile draft plasmid assemblies by providing a template for contig extension and gap closure when partial matches of plasmid typing loci map to the end of draft assembly contigs.

The Lociq method combines structural and sequence variants to increase the discriminatory power of existing plasmid typing methods. By reducing plasmids to their component parts, the Lociq method standardizes comparison metrics among plasmid types and allows for enhanced investigations between plasmid loci and plasmid metadata such as AMR gene composition, isolation source or plasmid lineage. The results of the Lociq method will not only benefit basic plasmid biology research they will also aid public health monitoring programs such as NARMS to track the spread of plasmid lineages and better identify the origin of multi-drug resistant plasmids.

## Methods

**Sequences and core annotations**. The initial dataset of long read sequences from 175 *Salmonella* and *E. coli* retail meat and cecal sample NARMS isolates were generated using PacBio Sequel platform with sequencing kit v3.0 (Pacific Biosciences, Menlo Park, CA). Sequencing libraries were prepared with the PacBio SMRTbell template prep kit v1.0 and the resulting reads were assembled into closed contigs using the PacBio Hierarchical Genome Assembly Process 4.0 and Circlator v1.5.5[28,29]. Plasmid Inc type was determined using PlasmidFinder definitions (accessed 4-27-2022) and closed plasmid assemblies were annotated with PROKKA v1.14.5[30,31]. The reference database of plasmid sequences evaluated was the PLSDB database v. 2021_06_23_v2[24].

**Lociq method**. The scripts for operation of the Lociq method are available for download at http://www.github.com/LBHarrison/Lociq/. Required input for the method includes annotation files of closed plasmid assemblies, access to reference database and plasmid type metadata. Additionally, the program requires user defined thresholds for prevalence and distance to account for variability in diversity among different plasmid groups and comparator schema.

**Identification of plasmid typing Loci**. The pangenome and intergenic regions of the closed plasmid dataset were obtained using Roary and piggy, respectively[23,22]. Data for the coding and intergenic pangenomes were merged and passed to R for clustering as binary data with complete linkage[32]. Plasmid typing loci among the Inc groups was determined in a two-stage process (Fig. 1). First, putative plasmid typing loci were identified by selecting the loci with a user-defined threshold of high prevalence in the plasmid group of interest and a user-defined threshold of low prevalence in the other plasmid groups. Second, loci were queried against an external plasmid database as a validation step using an 80% identity threshold. Loci that met or exceeded user-defined prevalence thresholds for membership within a plasmid group were identified as the plasmid typing loci for the current plasmid group.

**Identification of conserved plasmid fragments**. Sequence coordinates of typing loci were obtained through a BLAST query of the loci against an external plasmid database. Clusters of loci separated by less than a user-defined threshold value defined the contiguous sequence regions of a plasmid. These data were used to generate a contingency table displaying an all vs all tally of loci occurring in the same contiguous sequence region. The contingency table was analyzed as a correlation matrix evaluating the Pearson's correlation coefficient (R) for all loci interactions using the R Hmisc v4.7-0 package[33]. Loci clusters with a mean correlation coefficient ≥ 0.9 represent conserved contiguous sequence regions in the plasmid dataset and are referred to as plasmid fragment. Loci clusters with a mean R-value < 0.9 were subjected to increasingly stringent clustering parameters until the resulting plasmid fragments had a mean R-value ≥ 0.9.

**Plasmid subtyping**. Plasmid sequences were indexed to begin at the typing locus present in the greatest number of plasmids and the sequences were analyzed with AMRFinder plus to identify AMR genes and stress tolerance genes[34]. Plasmids were then subtyped using the metrics of: sequence type, organization of loci, organization of plasmid fragments and the distances between the plasmid fragments (Fig. 4). Unique numeric identifiers of the typing metrics are generated as part of the summary file output from the Lociq program (Supplementary Data 4). Plasmid sequence type was determined by the allelic composition of plasmid typing loci. Loci position data were extracted from the BLAST results and a unique numeric identifier was assigned to each unique organization of loci among the plasmid typing fragments. A similar process was applied to the order of plasmid fragments to determine the plasmid fragment type. Finally, the distances between each plasmid fragment were recorded to identify each plasmid's set of interfragment distance values.

**Comparator typing methods**. Comparator typing schema were used to evaluate the discriminatory power of the Lociq method. PTU designation and MOB type were determined using COPLA (updated 6-30-2021 using the RS84 definitions)[19]. IncC plasmids were further characterized by the IncA/C PMLST and IncA/C cgPMLST allelic profiles as recorded in PubMLST (Accessed 8-18-2022)[3,35]. Discriminatory power of the typing schema was determined by Simpson's diversity index.

**Downstream analyses**. Downstream analyses were performed to demonstrate four additional applications of Lociq method. In the first demonstration of custom annotations, insertion sequence (IS) elements were identified in the dataset using ISEscan v1.7.23 and the results merged with the Lociq annotation file[36]. Sequence alignments were generated with NCBI BLAST and visualizations were generated using the R genoPlotR package[37]. In the second demonstration that identified trends in the position of AMR genes in the dataset, the distance of an AMR gene of interest to its nearest plasmid fragments were visualized on a density plot in the base R package.

Third, the Lociq method was used to improve characterization of plasmid draft assemblies. This was done by performing a BLAST query of typing loci against the draft plasmid assembly to identify loci present in the sequence. The results were filtered by requiring an identity > 70% and coverage >90%. The draft assembly loci sequences were compared to the reference plasmid loci sequences to determine which specific plasmid typing alleles were present in the draft assembly. Alleles present in the draft plasmid assembly were used to construct an $m \times n$ presence/absence matrix where $m$ was equal the number of plasmids that were analyzed with the Lociq method + the plasmid draft assembly and $n$ was equal to number of unique alleles in the plasmid draft assembly. The presence/absence matrix was used to create a distance matrix of plasmids using the dist function in R with the method parameter set to binary. The row corresponding to the plasmid draft assembly was extracted and the distance values were evaluated to identify the least dissimilar plasmids from the Lociq results.

**Statistics and reproducibility**. Statistical tests were performed in R with the Hmsic package[32,33]. Specifically, the Pearson's correlation coefficient and corresponding probabilities were calculated to evaluate the pairwise likelihood of any

two plasmid typing loci occurring on the same region of the plasmid. Source data and numeric results are available in Supplementary Data 3. The sample size for these tests accounted for the 72 plasmid typing loci that were identified in the IncC plasmid demonstration dataset.

**Reporting summary**. Further information on research design is available in the Nature Portfolio Reporting Summary linked to this article.

## Data availability

Plasmid sequences are available through the PLSDB database (https://ccb-microbe.cs.uni-saarland.de/plsdb/) while plasmid metadata files are hosted in a Github repository (https://github.com/LBHarrison/Lociq/). Source data for Figs. 3, 6 and 8 are provided in Supplementary Data 3, 7 and 8, respectively.

## Code availability

The Lociq program is available through the Github repository https://github.com/LBHarrison/Lociq/[38].

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

## Acknowledgements

This work was supported through the U.S. FDA National Antimicrobial Resistance Monitoring System. The views expressed in this paper are those of the authors and may not reflect the official policy of the FDA, the Department of Health and Human Services, or the U.S. Government. Reference to any commercial materials, equipment, or process does not in any way constitute approval, endorsement, or recommendation by the FDA.

## Author contributions

C.L., E.S., G.H.T., L.H., P.F.M., and S.Z. contributed to data analysis & interpretation as well as manuscript revision. G.H.T., L.H. and S.Z. were responsible for conception and design of the project while SZ was responsible for project oversight. L.H. was responsible for the figure and software creation as well as the draft manuscript. C.L. was responsible for data acquisition. Funding was obtained by P.F.M. and S.Z.

## Competing interests

The authors declare no competing interests.
