## [Peer Review File · Communications Biology]

Reviewers' comments:

Reviewer #1 (Remarks to the Author):

Overall, the authors present a novel method for characterizing plasmids, and it is a much needed contribution to the field. The section that describes the evolution of closely related *Klebsiella* plasmids highlights the advance this method can provide. In addition, the authors' willingness to consider non-closed plasmid sequence (draft genomes) is a critical feature of the Lociq method. Lociq also appears to provide far more fine-scale resolution than current methods, and I hope that this paper will be published.

That said, I think there are some changes that could (and should) be made to improve the paper (these are general comments; more specific comments are at the end of the review):

1. The process by which the typing scheme for IncC was developed would benefit greatly from a schematic diagram. While there is a diagram for the general case (which is very good), understanding the method and the results would benefit greatly from moving this figure forward to the results (or perhaps even the introduction?).

2. Regarding the methods, the authors state that an 80% identity threshold for validating clustered loci was used (p. 27). What is the justification for that threshold and does locus assignment differ if the identity threshold is altered?

3. The reader is faced with many terms (e.g., loci pattern) that could be misinterpreted or misunderstood. The authors should define these terms explicitly in the manuscript and should ensure that they are used consistently throughout the manuscript (including the supplemental tables and figures) E.g., in line 188, what is "fragment type"?

4. One thing that was not clear in the paper (though I might have misunderstood one of the tables) is exactly what the output of Lociq is (is it Table 1 or S4?). There should be a clearer description in the text (perhaps the methods section).

5. An advantage of an advantage of MLST approaches is their portability (which also makes them easier to implement at a high-throughput data pipelines) since MLST alleles are fixed entities (i.e., an MLST locus is a defined sequence and number and does not change). With Lociq, it appears that the loci could change depending on the data used to identify loci, which means different users could develop different typing schemes. How would the authors limit (or possibly prevent this)?

6. How robust are the loci to the loss of sequences? That is, if one or two sequences were removed from the locus creation step, does that change the selection of loci?

7. Related to the previous point, what would be the authors' suggestions for implementation in high-throughput pipelines? In terms of surveillance, we would not want different surveillance organizations to have different typing schemes. In terms of implementation, how much recomputing could be necessary for large-scale implemented (i.e., will `fragment_pattern`, `loci_pattern`, `plasmid_ST`) have to be recalculated as new data arrive? If so, how frequently etc.?).

8. How is the identification of AMR loci performed? Is this just pulled from the PROKKA annotation?

Specific comments:

1. line 45: "the plasmids" could read "plasmids"
2. line 130: what clustering algorithm/method was used?

Reviewer #2 (Remarks to the Author):

Harrison et al present a bioinformatics tool Lociq which provides a new way of classifying plasmids using high-quality assemblies which have become increasingly common through high-quality long-read sequencing technologies such as PacBio HiFi. This new classification, provided by Lociq, is a highly relevant addition to our understanding of plasmid diversity, first due to the high biomedical importance of plasmids (eg. their capacity to carry antibiotic resistance genes), as well as plasmids' inherent capability for mobility and recombination.

The manuscript is well written, providing an interesting introduction where the characteristics and shortcomings of the previous plasmid classification concepts are discussed, after which the motivation for a tool like Lociq becomes obvious. Focusing on the IncC subclass of plasmids is well justified, although possibly more attention could be added (particularly in the web resources) for facilitating the end users to add new plasmid subclassed. The software is described clearly, and the provided use cases are interesting and highly relevant with obvious clinical implications. I particularly liked the concept of standardized language for plasmid feature diversity.

The method itself is an impressive software package which includes a graphical Shiny app for visual exploration of the results. Unfortunately, I did have issues with the Conda installation (specific to Conda itself rather than Lociq) so my suggestion would be to wrap the software into eg. a Docker or Singularity container while also keeping the Conda installation option.

I find it somewhat unusual that the Discussion part of the manuscript provides no references to literature – however, in this case I consider this acceptable since the introduction is comprehensive and provides a thorough introduction to plasmid classification methods and their shortcomings.

I'm pleased to recommend accepting the manuscript without further revisions. I would however advise the authors to improve the web resources by providing clearer examples of adding plasmid groups other than IncC, and by adding the possibilities to run Lociq as a Docker or Singularity container.

Reviewer #3 (Remarks to the Author):

The authors describe in this manuscript a novel bioinformatic tool and a framework to leverage sequence information for typing complete plasmids. As clearly stated by the authors: « The purpose of this paper is to introduce the plasmid subtyping method, demonstrate its ability to subtype IncC plasmids, compare it to existing plasmid typing methods, and show how the results of the subtyping ». And then proceeds with presenting very convincingly the results and demonstrates how the tool can be used to precisely type plasmids from sequence alone. I think the presented tool and framework is addressing a very much needed improvement of plasmid typing using sequence information alone from nowadays, thanks to long reads, relatively easy to obtain complete plasmid sequences. A lot of efforts are made to make the framework robust and consistent in naming, which has the potential to be a reference system.

The paper is well written and convincing, and I would recommend it for publication after some minor adjustments and suggestions.

A first thing that struck me is the absence of any analysis of plasmid backbone phylogeny is incorporated, although it seems quite easy to leverage. It would then be very powerful to compare trees of backbone genes evolution, but this is probably beyond the scope of a typing method.

Another point not mentioned by authors but where I see very high potential is the typing of non-archetypal plasmids, for example obtained from environmental shotgun assemblies. With long reads sequencing, more and more complete plasmids can be assembled that have no close relative in current databases. This method could prove useful to organize these new sequences as well.

The journal format request methods to be put at the end, but it would be helpful in Results to very briefly introduce some terms used throughout. For example, it is unclear how « loci » are defined bioinformatically. It could be helpful to say that loci are here defined as the combined output of Roary & Piggy, as I understood from the method.

Figure1 : The rows (loci) clustering does not convey a lot of information in itself and could be removed as clustering of marker loci for each closed plasmid Inc group is obvious ?

L150-153 – Unsure I understand. Did you compare all IncC typed versus all non-typed plasmids in PLSDB or against all not typed as IncC in PLSDB ?

Figure 6: From the legend alone I don't understand how the relative position of bla_{CMY-2} is represented as the x-axis is an absolute distance from the start of each fragment (?).

L287-311: Although there is no factual argument contradicting the authors assumption on the sequence of events described, there are no factual arguments suggesting their assumption is true either. Although unfamiliar with the study that performed this longitudinal sampling, it is likely that the sequences are obtained from isolates. There is no evidence that the plasmid isolated and sequenced in 2015 was not already existing, but not sampled, in 2013. If epidemiological data on plasmids are available in the original paper, briefly cite these results. If not, I would rephrase to clearly state this is a plausible scenario rather than an acquired fact, that is anyway beyond the point of this paper. It would also be interesting to incorporate some phylogeny, based on conserved backbone genes together with the analysis of lateral transfers and recombination events

Original:

Referee expertise:

Referee #1: AMR gene prediction

Referee #2: AMR genomics, bioinformatics

Referee #3: Metagenomics, AMR monitoring

Reviewers' comments:

Reviewer #1 (Remarks to the Author):

Overall, the authors present a novel method for characterizing plasmids, and it is a much needed contribution to the field. The section that describes the evolution of closely related *Klebsiella* plasmids highlights the advance this method can provide. In addition, the authors' willingness to consider non-closed plasmid sequence (draft genomes) is a critical feature of the Lociq method. Lociq also appears to provide far more fine-scale resolution than current methods, and I hope that this paper will be published.

That said, I think there are some changes that could (and should) be made to improve the paper (these are general comments; more specific comments are at the end of the review):

1. The process by which the typing scheme for IncC was developed would benefit greatly from a schematic diagram. While there is a diagram for the general case (which is very good), understanding the method and the results would benefit greatly from moving this figure forward to the results (or perhaps even the introduction?).
2. Regarding the methods, the authors state that an 80% identity threshold for validating clustered loci was used (p. 27). What is the justification for that threshold and does locus assignment differ if the identity threshold is altered?
3. The reader is faced with many terms (e.g., loci pattern) that could be misinterpreted or misunderstood. The authors should define these terms explicitly in the manuscript and should ensure that they are used consistently throughout the manuscript (including the supplemental tables and figures) E.g., in line 188, what is "fragment type"?
4. One thing that was not clear in the paper (though I might have misunderstood one of the tables) is exactly what the output of Lociq is (is it Table 1 or S4?). There should be a clearer description in the text (perhaps the methods section).
5. An advantage of an advantage of MLST approaches is their portability (which also makes them easier

to implement at a high-throughput data pipelines) since MLST alleles are fixed entities (i.e., an MLST locus is a defined sequence and number and does not change). With Lociq, it appears that the loci could change depending on the data used to identify loci, which means different users could develop different typing schemes. How would the authors limit (or possibly prevent this)?

6. How robust are the loci to the loss of sequences? That is, if one or two sequences were removed from the locus creation step, does that change the selection of loci?

7. Related to the previous point, what would be the authors' suggestions for implementation in high-throughput pipelines? In terms of surveillance, we would not want different surveillance organizations to have different typing schemes. In terms of implementation, how much recomputing could be necessary for large-scale implemented (i.e., will `fragment_pattern`, `loci_pattern`, `plasmid_ST`) have to be recalculated as new data arrive? If so, how frequently etc.?).

8. How is the identification of AMR loci performed? Is this just pulled from the PROKKA annotation?

Specific comments:

1. line 45: "the plasmids" could read "plasmids"
2. line 130: what clustering algorithm/method was used?

Reviewer #2 (Remarks to the Author):

Harrison et al present a bioinformatics tool Lociq which provides a new way of classifying plasmids using high-quality assemblies which have become increasingly common through high-quality long-read sequencing technologies such as PacBio HiFi. This new classification, provided by Lociq, is a highly relevant addition to our understanding of plasmid diversity, first due to the high biomedical importance of plasmids (eg. their capacity to carry antibiotic resistance genes), as well as plasmids' inherent capability for mobility and recombination.

The manuscript is well written, providing an interesting introduction where the characteristics and shortcomings of the previous plasmid classification concepts are discussed, after which the motivation for a tool like Lociq becomes obvious. Focusing on the IncC subclass of plasmids is well justified, although possibly more attention could be added (particularly in the web resources) for facilitating the end users to add new plasmid subclassed. The software is described clearly, and the provided use cases are interesting and highly relevant with obvious clinical implications. I particularly liked the concept of standardized language for plasmid feature diversity.

The method itself is an impressive software package which includes a graphical Shiny app for visual exploration of the results. Unfortunately, I did have issues with the Conda installation (specific to Conda itself rather than Lociq) so my suggestion would be to wrap the software into eg. a Docker or Singularity container while also keeping the Conda installation option.

I find it somewhat unusual that the Discussion part of the manuscript provides no references to literature – however, in this case I consider this acceptable since the introduction is comprehensive and provides a thorough introduction to plasmid classification methods and their shortcomings.

I'm pleased to recommend accepting the manuscript without further revisions. I would however advise the authors to improve the web resources by providing clearer examples of adding plasmid groups other than IncC, and by adding the possibilities to run Lociq as a Docker or Singularity container.

Reviewer #3 (Remarks to the Author):

The authors describe in this manuscript a novel bioinformatic tool and a framework to leverage sequence information for typing complete plasmids. As clearly stated by the authors: « The purpose of this paper is to introduce the plasmid subtyping method, demonstrate its ability to subtype IncC plasmids, compare it to existing plasmid typing methods, and show how the results of the subtyping ». And then proceeds with presenting very convincingly the results and demonstrates how the tool can be used to precisely type plasmids from sequence alone. I think the presented tool and framework is addressing a very much needed improvement of plasmid typing using sequence information alone from nowadays, thanks to long reads, relatively easy to obtain complete plasmid sequences. A lot of efforts are made to make the framework robust and consistent in naming, which has the potential to be a reference system.

The paper is well written and convincing, and I would recommend it for publication after some minor adjustments and suggestions.

A first thing that struck me is the absence of any analysis of plasmid backbone phylogeny is incorporated, although it seems quite easy to leverage. It would then be very powerful to compare trees of backbone genes evolution, but this is probably beyond the scope of a typing method.

Another point not mentioned by authors but where I see very high potential is the typing of non-archetypal plasmids, for example obtained from environmental shotgun assemblies. With long reads sequencing, more and more complete plasmids can be assembled that have no close relative in current databases. This method could prove useful to organize these new sequences as well.

The journal format request methods to be put at the end, but it would be helpful in Results to very briefly introduce some terms used throughout. For example, it is unclear how « loci » are defined bioinformatically. It could be helpful to say that loci are here defined as the combined output of Roary & Piggy, as I understood from the method.

Figure1 : The rows (loci) clustering does not convey a lot of information in itself and could be removed as clustering of marker loci for each closed plasmid Inc group is obvious ?

L150-153 – Unsure I understand. Did you compare all IncC typed versus all non-typed plasmids in PLSDB or against all not typed as IncC in PLSDB ?

Figure 6: From the legend alone I don't understand how the relative position of blaCMY-2 is represented as the x-axis is an absolute distance from the start of each fragment (?).

L287-311: Although there is no factual argument contradicting the authors assumption on the sequence of events described, there are no factual arguments suggesting their assumption is true either. Although unfamiliar with the study that performed this longitudinal sampling, it is likely that the sequences are obtained from isolates. There is no evidence that the plasmid isolated and sequenced in 2015 was not already existing, but not sampled, in 2013. If epidemiological data on plasmids are available in the original paper, briefly cite these results. If not, I would rephrase to clearly state this is a plausible scenario rather than an acquired fact, that is anyway beyond the point of this paper. It would also be interesting to incorporate some phylogeny, based on conserved backbone genes together with the analysis of lateral transfers and recombination events.

Responses:

Reviewers' comments:

Reviewer #1 (Remarks to the Author):

Overall, the authors present a novel method for characterizing plasmids, and it is a much needed contribution to the field. The section that describes the evolution of closely related *Klebsiella* plasmids highlights the advance this method can provide. In addition, the authors' willingness to consider non-closed plasmid sequence (draft genomes) is a critical feature of the Lociq method. Lociq also appears to provide far more fine-scale resolution than current methods, and I hope that this paper will be published.

Response: We appreciate the reviewer's positive feedback and share in the hope that this paper will be published.

That said, I think there are some changes that could (and should) be made to improve the paper (these are general comments; more specific comments are at the end of the review):

Q1. The process by which the typing scheme for IncC was developed would benefit greatly from a schematic diagram. While there is a diagram for the general case (which is very good), understanding the method and the results would benefit greatly from moving this figure forward to the results (or perhaps even the introduction?).

Response 1. We agree with the reviewer regarding the order of material presentation and have moved Figures 8 & 9 to the corresponding sections of the Results. This way, the IncC-specific results are presented in the context of how plasmid definitions are generated in a more general case.

Q2. Regarding the methods, the authors state that an 80% identity threshold for validating clustered loci was used (p. 27). What is the justification for that threshold and does locus assignment differ if the identity threshold is altered?

Response 2. We selected the permissive 80% threshold value to account for the sequence diversity within plasmid loci in the reference database. Fortunately, the 80% identity threshold is only the first filter for loci validation. The user-defined second and third filters (being A – the prevalence within the group of interest and B - the prevalence outside the group of interest) serve to remove the subset of loci

with high identity that are not well represented within the plasmid group, or over-represented outside the plasmid group.

We believe that alterations in the percent identity threshold will be plasmid-type specific and will follow the trend of an increasing percent identity threshold corresponding to fewer typing loci being recovered. To test this, we evaluated different stringency thresholds in our IncC and IncFIB(pN55391) plasmid groups.

Within our evaluation of the IncC plasmids, the lowest percent identity match was 81.14, so we were unable to evaluate the effects of decreasing the threshold values. Increasing the stringency to 85% & 90% identity had no effect on loci selection (72 were recovered), however 95% and 99% identity thresholds resulted in 68 & 43 loci, respectively.

We pursued this further with the IncFIB(pN55391) plasmids where the lowest loci percent identity match was 70.68. Reducing the threshold from 80% to 70% resulted in an 11% increase in the number of loci recovered. This is consistent with the dynamic that the number of recovered loci increases with decreasing percent identity stringency thresholds.

Q3. The reader is faced with many terms (e.g., loci pattern) that could be misinterpreted or misunderstood. The authors should define these terms explicitly in the manuscript and should ensure that they are used consistently throughout the manuscript (including the supplemental tables and figures) E.g., in line 188, what is "fragment type"?

Response 3. We agree that the terms could have been defined more clearly and at more appropriate times in the manuscript. We have added the following sentences to the Results section (Lines 181-189) that addresses the typing of IncC plasmids from the external database:

Plasmid characterization was performed by assigning a numeric identifier to each unique pattern of sequence type, fragment type and loci type (Figure Methods 2). The plasmid sequence type was defined by the complement of plasmid alleles in the plasmid, regardless of their position. The plasmid fragment type was determined by how the plasmid fragments were ordered along the plasmid, relative to a semi-conservative starting locus. The plasmid loci type was determined by rearranging the plasmid fragments in ascending order of their numeric identifier and recovering the arrangement of loci from the re-ordered plasmid fragments. This temporary rearrangement of plasmid fragments for loci typing allows the loci type to be independent of the fragment type.

Note- the semi-conservative indexing locus is referenced in the Methods section under Plasmid Subtyping.

Q4. One thing that was not clear in the paper (though I might have misunderstood one of the tables) is exactly what the output of Lociq is (is it Table 1 or S4?). There should be a clearer description in the text (perhaps the methods section).

Response 4. The reviewer is correct that the primary output for Lociq is Table S4. This table contains the plasmid ID, sequence type, loci pattern, fragment pattern and interfragment distances. Table 1 is a subset of Table S4 that we modified to fit page dimensions.

However, depending on the stage of analysis, many outputs are generated. If Lociq is used to characterize new plasmid sequences against an existing set of plasmid definitions, it will produce:

- Table detailing the summary of typing results (This is table S4)

- Multifasta file of indexed plasmid sequences

- Updated Table detailing the AMR results for all plasmids analyzed

- Updated reference tables containing the sequence definitions of all loci/allele combinations, loci patterns and fragment patterns.

Further, if the user is running the entire Lociq program to establish definitions for a new plasmid type, the following will be generated:

- Image of a presence/absence matrix of the plasmid pangenome (Figure 2, without the colored bars)

- Image of a presence/absence matrix of the plasmid pangenome with an indicator of the plasmid group of interest (Figure 2, with only the indicator bars for the plasmid type of interest)

- Loci prevalence plots both before and after validation against an external database (Figure 1: upper right and lower middle-left)

- Multifasta files of the pre- and post-validation plasmid typing loci

- Image of the correlogram of plasmid fragments (Figure 3)

- Table detailing the R & p values for plasmid fragment correlation coefficients (Table S3)

- Plasmid typing loci used to index the start position of plasmid sequences

- Reference file detailing which locus belongs to which fragment

- Table detailing the summary of typing results (This is table S4)

- Multifasta file of indexed plasmid sequences

- Table detailing the AMR results for all plasmids analyzed

- Reference tables containing the sequence definitions of all loci/allele combinations, loci patterns and fragment patterns.

Additionally, there is an R Shiny application for results exploration. (Images similar to Figure 3, though figure annotation text may overlap)

We have listed the primary outputs that are generated in lines 200-212, however the importance of each output will be dependent on the end-user's application.

While we do not have the space in the manuscript to include all of the outputs generated while using Lociq, we have updated the Github repository to include a detailed description of each step, including the expected outputs. (<https://github.com/LBHarrison/Lociq#program-details>).

Q5. An advantage of an advantage of MLST approaches is their portability (which also makes them easier to implement at a high-throughput data pipelines) since MLST alleles are fixed entities (i.e., an MLST locus is a defined sequence and number and does not change). With Lociq, it appears that the loci could change depending on the data used to identify loci, which means different users could develop different typing schemes. How would the authors limit (or possibly prevent this)?

Response 5. Without standardization, the definitions are limited to in-house research and data exploration applications. For plasmid typing methods to become standardized, the reviewer is absolutely correct that users would need access to the same starting dataset, or access to a common set of definitions.

One of our goals is to foster collaboration between researchers to identify a suitably diverse and representative dataset from which to generate these plasmid definitions and create a publicly available reference database. We believe that submission and revision of the manuscript and Lociq method for peer review is a critical first step in this pursuit.

Q6. How robust are the loci to the loss of sequences? That is, if one or two sequences were removed from the locus creation step, does that change the selection of loci?

Response 6. The answer to this question is influenced by two variables: 1) what is the prevalence of the typing loci among the plasmid sequences, and 2) how many sequences of a given plasmid type are present in the initial dataset of long-read sequences.

First, if prevalence of a specific locus among the plasmid group of interest is 100%, then the removal of any one of the plasmid sequences will not affect the prevalence because the loci will still be present in all of the remaining sequences. If the prevalence is below 100%, then the matter becomes dependent on the size of the initial starting set.

With a large starting dataset and high prevalence of a locus among a specific plasmid type, the effects of sequence loss will be minimal. However, as sample size decreases and the locus prevalence approaches the user-defined prevalence threshold value, then a loss of sequence will have a greater likelihood of decreasing the overall prevalence of the locus in the typing group and assigning it below the user-defined threshold value.

In short, the loci with the lowest prevalence in a plasmid group of interest (and therefore the least representative) will be disproportionately affected by a decreasing number of sequences in the initial dataset.

This highlights the importance of validating a dataset against a large external database. In the unexpected event that the prevalence of a low-frequency typing-locus is overrepresented in a small

starting dataset, its decreased prevalence in the external dataset would address the initial overrepresentation and the locus would be removed from the set of typing loci.

Q7. Related to the previous point, what would be the authors' suggestions for implementation in high-throughput pipelines? In terms of surveillance, we would not want different surveillance organizations to have different typing schemes. In terms of implementation, how much recomputing could be necessary for large-scale implemented (i.e., will fragment_pattern, loci_pattern, plasmid_ST) have to be recalculated as new data arrive? If so, how frequently etc.?).

Response 7. The reviewer raises an excellent question and a centralized database of plasmid definitions would be ideal for high-throughput pipelines. In its current state, the program is designed to match new plasmid sequences to existing plasmid definitions first, and only perform the analysis to develop new numeric IDs for fragment, loci and sequence variations if novel variants are discovered.

Since the numeric identifiers for sequence type, loci pattern and fragment pattern represent specific sequences and patterns, the addition of new sequences or patterns to the database will not affect the existing data entries. For example, an IncC 30.151.84 plasmid will remain IncC 30.151.84 regardless of how many IncC plasmid variants are discovered.

We speculate that updating a centralized database would take part in two stages. The first stage would be a simple update that adds new plasmid definitions as novel patterns are identified by end-users. This process would only be slightly more complicated than adding a row to a reference table and would require minimal computing resources. The second stage would only be necessary if there were a fundamental shift in our understanding of the composition of a given plasmid type. Likely, this would be an infrequent occurrence. Updating this could be performed centrally to minimize computational requirements, then the updated definition reference files could be distributed. As the initial plasmid definitions covered in the manuscript were generated on a desktop computer (Ubuntu 20.04, AMD Ryzen 7-3700x, 64Gb RAM), these infrequent updates should not strain the computational resources of monitoring organizations.

Q8. How is the identification of AMR loci performed? Is this just pulled from the PROKKA annotation?

Response 8. The results are extracted from the output of AMRFinderPlus. We recognize the reviewer's concern and have added a citation for AMRFinder plus to the new location at line 469-470.

Specific comments:

Specific Q1. line 45: "the plasmids" could read "plasmids"

Specific Response 1. We agree with the reviewer and have changed lines 44-45 from:

This shortfall of using a single target for plasmid typing is apparent when the plasmids contain multiple replicon sequences.

to

This shortfall of using a single target for plasmid typing is apparent when plasmids contain multiple replicon sequences.

Specific Q2. line 130: what clustering algorithm/method was used?

Specific Response 2. We agree with the reviewer that the reader should be aware of the method when it is first introduced.

We have added the following sentences to the paragraph in lines 136-140:

The pangenome was analyzed as a presence/absence matrix (PAM) in R where plasmids were grouped by the similarity of their loci profiles accounting for both the coding and intergenic regions. This grouping was performed first by computing a distance matrix of the binary PAM data, then clustering with the hclust function using complete linkage.

Reviewer #2 (Remarks to the Author):

Harrison et al present a bioinformatics tool Lociq which provides a new way of classifying plasmids using high-quality assemblies which have become increasingly common through high-quality long-read sequencing technologies such as PacBio HiFi. This new classification, provided by Lociq, is a highly relevant addition to our understanding of plasmid diversity, first due to the high biomedical importance of plasmids (eg. their capacity to carry antibiotic resistance genes), as well as plasmids' inherent capability for mobility and recombination.

The manuscript is well written, providing an interesting introduction where the characteristics and shortcomings of the previous plasmid classification concepts are discussed, after which the motivation for a tool like Lociq becomes obvious. Focusing on the IncC subclass of plasmids is well justified, although possibly more attention could be added (particularly in the web resources) for facilitating the end users to add new plasmid subclassed. The software is described clearly, and the provided use cases are interesting and highly relevant with obvious clinical implications. I particularly liked the concept of standardized language for plasmid feature diversity.

The method itself is an impressive software package which includes a graphical Shiny app for visual exploration of the results. Unfortunately, I did have issues with the Conda installation (specific to Conda itself rather than Lociq) so my suggestion would be to wrap the software into eg. a Docker or Singularity container while also keeping the Conda installation option.

I find it somewhat unusual that the Discussion part of the manuscript provides no references to literature – however, in this case I consider this acceptable since the introduction is comprehensive and provides a thorough introduction to plasmid classification methods and their shortcomings.

I'm pleased to recommend accepting the manuscript without further revisions. I would however advise the authors to improve the web resources by providing clearer examples of adding plasmid groups other than IncC, and by adding the possibilities to run Lociq as a Docker or Singularity container.

Response: First, we thank the reviewer for their positive feedback.

We agree with the reviewer that the web resources could be improved to make the program more accessible. We have created a Dockerized version of Lociq that can be installed from https://github.com/LBHarrison/Lociq_docker. The link to the Docker version of Lociq is available through the primary Lociq link (<https://github.com/LBHarrison/Lociq>). This has the added benefit of allowing the program to be run on WSL-enabled Windows platforms.

We have also added a section to the Github repository that describes how the user may analyze other plasmid types. This requires the user to be familiar with how the reference metadata files are formatted, and we have included instructions to the repository on where the appropriate information may be found.

Reviewer #3 (Remarks to the Author):

The authors describe in this manuscript a novel bioinformatic tool and a framework to leverage sequence information for typing complete plasmids. As clearly stated by the authors: « The purpose of this paper is to introduce the plasmid subtyping method, demonstrate its ability to subtype IncC plasmids, compare it to existing plasmid typing methods, and show how the results of the subtyping ». And then proceeds with presenting very convincingly the results and demonstrates how the tool can be used to precisely type plasmids from sequence alone. I think the presented tool and framework is addressing a very much needed improvement of plasmid typing using sequence information alone from nowadays, thanks to long reads, relatively easy to obtain complete plasmid sequences. A lot of efforts are made to make the framework robust and consistent in naming, which has the potential to be a reference system.

The paper is well written and convincing, and I would recommend it for publication after some minor adjustments and suggestions.

Response: We appreciate the positive feedback from the reviewer

Q1. A first thing that struck me is the absence of any analysis of plasmid backbone phylogeny is incorporated, although it seems quite easy to leverage. It would then be very powerful to compare trees of backbone genes evolution, but this is probably beyond the scope of a typing method.

Response 1. We agree with the reviewer that this would be useful information for researchers and have included an accessory script to the GitHub repository that will allow the user to generate the phylogeny of the backbone of their current plasmid type. Additionally, we have included a dendrogram in line 171 following the initial characterization of IncC plasmids.

It is important to note, though, that the dendrogram was generated using the distance calculations from presence/absence matrix of typing loci/allele combinations. Unfortunately, we are unable to represent the phylogeny of a concatenated plasmid backbone using a maximum-likelihood tree. ML Trees require comparator sequences to be of equal length, however the concatenated typing-loci of our plasmids can have missing or even duplicated loci, introducing large variations in sequence length.

Q2. Another point not mentioned by authors but where I see very high potential is the typing of non-archetypal plasmids, for example obtained from environmental shotgun assemblies. With long reads sequencing, more and more complete plasmids can be assembled that have no close relative in current databases. This method could prove useful to organize these new sequences as well.

Response 2. We agree with the reviewer that one potential application of this method is the typing of non-archetypal plasmids. This is a concept we intended to address in the Discussion Lines 411-415, but did not convey properly. We have reworded the section to read:

First, the Lociq method can be used to characterize plasmids that are currently undefined and routinely under-studied. While plasmids from readily culturable bacteria have strong representation in plasmid classification databases, plasmids from slow growing and environmental samples may not be accounted for. Analysis of the plasmid pangenome through the Lociq method can address this research need because the program can be used to characterize plasmids independent of any external metadata. The initial stage of the Lociq program organizes plasmids independent of plasmid type through hierarchical clustering of loci presence/absence data. Plasmids belonging to clusters without a known plasmid type can be characterized by the Lociq method using the typing loci unique to that cluster.

Q3. The journal format request methods to be put at the end, but it would be helpful in Results to very briefly introduce some terms used throughout. For example, it is unclear how « loci » are defined bioinformatically. It could be helpful to say that loci are here defined as the combined output of Roary & Piggy, as I understood from the method.

Response 3. The reviewer is correct that the loci are the combined output of Roary and piggy and we agree that this should be made clear to the reader. We have altered the Methods section in Lines 133-136 to read:

The combined pangenome for all 459 plasmids contained 6,726 unique coding and intergenic regions, as generated by the Roary & piggy programs. These 6,726 genetic elements are the library of plasmid loci found among our plasmids.

Additionally, we recognize and agree with the reviewer's concern regarding the formatting. To address this, we have included a description of how the loci were defined in the Results section to inform the reader at the appropriate time in lines 122-125. The section now reads:

Identification of the typing loci was performed by using the Roary and piggy programs to define the pangenome of 459 closed plasmid sequences. Prevalence thresholds were used to determine which pangenomic loci were indicative of and exclusive to a given plasmid type. Finally, the candidate typing loci were validated against an external database (Figure 1).

Additionally, we moved the method descriptor Figures 8 & 9 from the Methods section to the Results section as the new Figures 1 & 4.

Q4. Figure1 : The rows (loci) clustering does not convey a lot of information in itself and could be removed as clustering of marker loci for each closed plasmid Inc group is obvious ?

Response 4. We agree with the reviewer and have removed the loci clustering dendrogram, though we would like to keep the plasmid dendrogram because it communicates some of the diversity of the untyped plasmids.

We have updated the figure to the following version:

Q5. L150-153 – Unsure I understand. Did you compare all IncC typed versus all non-typed plasmids in PLSDb or against all not typed as IncC in PLSDb ?

Response 5. We agree with the reviewer that statement was unclear and did not accurately reflect the process. To clarify, we compared the prevalence of typing loci among all plasmids typed as IncC in

PLSDB versus the prevalence of typing loci among all plasmids not typed as IncC in PLSDB. We have reworded the Lines 158-159 to read:

We compared the prevalence of typing loci between IncC and non-IncC plasmids in the database.

Q6. Figure 6: From the legend alone I don't understand how the relative position of blaCMY-2 is represented as the x-axis is an absolute distance from the start of each fragment (?).

Response 6. We agree with the reviewer that the use of the word 'relative' to refer to the absolute distance is incorrect. We have removed the term from the Figure 8 legend.

Q7. L287-311: Although there is no factual argument contradicting the authors assumption on the sequence of events described, there are no factual arguments suggesting their assumption is true either. Although unfamiliar with the study that performed this longitudinal sampling, it is likely that the sequences are obtained from isolates. There is no evidence that the plasmid isolated and sequenced in 2015 was not already existing, but not sampled, in 2013. If epidemiological data on plasmids are available in the original paper, briefly cite these results. If not, I would rephrase to clearly state this is a plausible scenario rather than an acquired fact, that is anyway beyond the point of this paper. It would also be interesting to incorporate some phylogeny, based on conserved backbone genes together with the analysis of lateral transfers and recombination events.

Response 7. We agree with the concern of the reviewer that sufficient epidemiological data were not cited from the reference publication to support the subset of findings we referenced. The sequence of events that we presented followed the model of transmission that was communicated in the reference paper. Our intent was not to perform a secondary analysis of their results, but to describe their results in the context of the Lociq analysis as a demonstration of how these data could be organized in similar plasmid-tracing studies.

To help express this, we have modified lines 306-308 to read:

To do this, we used the Lociq method to visualize the results of a study in a major hospital in Taiwan that tracked the transmission of blaOXA-48 from a plasmid to a *K. pneumoniae* chromosome over a three-year period.

And line 317-319 to read:

The primary study indicated that the first stage of plasmid evolution was observed between the plasmids NZ_CP040034.1 and NZ_CP040029.1 that were isolated in the first year of the sample period.

We agree with the reviewer that a phylogenetic analysis of the conserved backbone genes would be of interest to the reader in similar circumstances. However, in this example, the four plasmids discussed all belong to the same sequence type, so the sequences of their plasmid backbone genes are identical.

REVIEWERS' COMMENTS:

Reviewer #1 (Remarks to the Author):

The authors have addressed the criticisms I had of the previous submission. In particular, the manuscript reads much more clearly and is, in my opinion, easier for readers to follow. The methodological questions I raised have been addressed, and, where needed, added to the text: this will make it easier for others to understand and, if needed, reproduce the analyses (or parts of it). In addition, they have adequately dealt with the other reviewers' concerns.

I encourage the editor to accept this article for publication.

Reviewer #2 (Remarks to the Author):

With the inclusion of a Dockerized version of Lociq as well as the instructions for analysing plasmid types beyond IncC, I'm happy to recommend accepting the manuscript without further revisions.

Reviewer #3 (Remarks to the Author):

The authors very adequately addressed the few minor issues of their first submission raised by the three reviewers including my own.

I was already very positive about the tool and its need for the field, I am now completely satisfied with the supporting paper manuscript. I strongly support and recommend its publication. This is an excellent addition to sequence-only plasmid typing that is urgently needed before long-reads sequencing become routine and full, complete, closed plasmids are assembled without effort from any type of microbial isolates genomic survey.

I am now impatiently looking forward the implementation of a centralized database :)

My sincere congratulations for the work

REVIEWERS' COMMENTS:

Reviewer #1 (Remarks to the Author):

The authors have addressed the criticisms I had of the previous submission. In particular, the manuscript reads much more clearly and is, in my opinion, easier for readers to follow. The methodological questions I raised have been addressed, and, where needed, added to the text: this will make it easier for others to understand and, if needed, reproduce the analyses (or parts of it). In addition, they have adequately dealt with the other reviewers' concerns.

I encourage the editor to accept this article for publication.

Response to reviewer #1: We thank the reviewer for their efforts in improving the readability of article and the functionality of the method.

Reviewer #2 (Remarks to the Author):

With the inclusion of a Dockerized version of Lociq as well as the instructions for analyzing plasmid types beyond IncC, I'm happy to recommend accepting the manuscript without further revisions.

Response to reviewer #2: We thank the reviewer for their recommendations to expand the accessibility of the program.

Reviewer #3 (Remarks to the Author):

The authors very adequately addressed the few minor issues of their first submission raised by the three reviewers including my own.

I was already very positive about the tool and its need for the field, I am now completely satisfied with the supporting paper manuscript. I strongly support and recommend its publication. This is an excellent addition to sequence-only plasmid typing that is urgently needed before long-reads sequencing become routine and full, complete, closed plasmids are assembled without effort from any type of microbial isolates genomic survey.

I am now impatiently looking forward the implementation of a centralized database :)

My sincere congratulations for the work

Response to reviewer #3: We thank the reviewer for their logical review of our work and their comments that improved the communication of the results.